# Correlation of *Pseudomonas aeruginosa* Phage Resistance with the Numbers and Types of Antiphage Systems

**DOI:** 10.3390/ijms25031424

**Published:** 2024-01-24

**Authors:** Kevin A. Burke, Caitlin D. Urick, Nino Mzhavia, Mikeljon P. Nikolich, Andrey A. Filippov

**Affiliations:** Wound Infections Department, Bacterial Diseases Branch, Walter Reed Army Institute of Research, Silver Spring, MD 20910, USA; kevin.a.burke28.ctr@health.mil (K.A.B.); caitlin.d.urick.ctr@health.mil (C.D.U.); nino.mzhavia.ctr@health.mil (N.M.); mikeljon.p.nikolich.civ@health.mil (M.P.N.)

**Keywords:** *Pseudomonas aeruginosa*, phage therapy, phage resistance, PADLOC, antiphage systems, antiphage systems–phage resistance correlation

## Abstract

Phage therapeutics offer a potentially powerful approach for combating multidrug-resistant bacterial infections. However, to be effective, phage therapy must overcome existing and developing phage resistance. While phage cocktails can reduce this risk by targeting multiple receptors in a single therapeutic, bacteria have mechanisms of resistance beyond receptor modification. A rapidly growing body of knowledge describes a broad and varied arsenal of antiphage systems encoded by bacteria to counter phage infection. We sought to understand the types and frequencies of antiphage systems present in a highly diverse panel of *Pseudomonas aeruginosa* clinical isolates utilized to characterize novel antibacterials. Using the web-server tool PADLOC (prokaryotic antiviral defense locator), putative antiphage systems were identified in these *P. aeruginosa* clinical isolates based on sequence homology to a validated and curated catalog of known defense systems. Coupling this host bacterium sequence analysis with host range data for 70 phages, we observed a correlation between existing phage resistance and the presence of higher numbers of antiphage systems in bacterial genomes. We were also able to identify antiphage systems that were more prevalent in highly phage-resistant *P. aeruginosa* strains, suggesting their importance in conferring resistance.

## 1. Introduction

The coevolutionary history of bacteria and bacteriophages is ancient. While the exact origin of viruses is uncertain, a prominent hypothesis argues that viruses evolved from ancient cells before the last universal common ancestor of cellular life [1]. That would mean that the evolutionary history of phages and their bacterial hosts is as old or nearly as old as bacteria themselves, and it is often described as an “arms race”. Bacteria frequently alter or hide phage receptors on the cell surface to evade detection by phages; these are often lipopolysaccharides (LPS) (for gram-negative bacteria), capsule, or various surface-localized proteins. In response, selection for mutations in receptor-binding proteins (RBP) enables phages to use modified or alternative receptors [2,3,4]. But this is not the full picture. Bacteria and phages have also developed numerous other active measures to counter phage predation and subsequently counter the resultant antiphage defenses. In recent years, dozens of distinct antiphage systems have been identified and characterized [5,6,7,8,9,10,11,12,13,14,15,16,17], adding to long-known mechanisms that counter phage infection, such as restriction modification and CRISPR-Cas systems [18]. These systems can work synergistically, indicating the importance of a more comprehensive understanding of bacterial antiphage strategies and how they work together. Given the abundance and diversity of phages and their hosts, a complete picture is only likely to emerge far in the future. Phage-defense genes are frequently grouped into “islands” in bacterial genomes, and sequence analysis has found that uncharacterized adjacent genes are often colocalized with these islands, suggesting that many additional antiphage genes are yet to be discovered [19]. Beyond distinct, dedicated antiphage enzymes, bacteria also employ broader strategies, including nutrient depletion or the production of small molecules with antiphage properties [20,21,22,23,24]. Taken together, the current body of knowledge provides a preliminary picture of what is likely a profoundly vast and varied array of antiphage mechanisms that can work synergistically to inhibit or block phage predation.

Conversely, phages have coevolved to counter bacterial mechanisms that reduce or block infection. In addition to RBP modification to adsorb to the cell surface to facilitate infection [25], phages are also known to alter CRISPR protospacer adjacent motif (PAM) sequences, restriction recognition sites, or other targets of antiphage systems [26,27]. More recently, it has become clear that phages are actively countering antiphage systems, producing enzymes that neutralize bacterial antiphage enzymes in a variety of ways [28]. The discovery of these phage antidefense (“anti-antiphage”) enzymes is rapidly emerging with improved sequencing and analysis of phage genes. Efforts to identify phage antidefense systems are made difficult by the large proportion of phage genomes that remain uncharacterized and of gene-encoding products of unknown function. However, recent work to compile known antiphage systems into catalogs is improving efforts at discovery [29]. It is expected that more antidefense mechanisms will be discovered with the current intensity of phage genomic and genetic studies.

Phages have been proposed for use in both therapeutic and industrial contexts, starting almost immediately after their discovery [30,31,32]. Successful application of phages in human compassionate-treatment cases has been widely reported, and there are ongoing and future clinical trials on phage therapy against drug-resistant infections [33,34,35]. While early efforts show promise, the development of effective, durable phage therapeutics will require the reduction of the emergence of phage resistance during treatment. So far, efforts have focused on using multiple phages in a cocktail that targets different surface receptors [36,37] because multiple receptors have to be altered to overcome the activity of the cocktail. Also, phage-resistant mutants that arise under the pressure of multiple phages using different receptors tend to be less fit or viable because mutations removing or modifying cell-surface molecules often come at a fitness cost for the bacterial cell, and altering two different receptors at once can have a much steeper cost.

However, focusing solely on surface receptors ignores the growing picture that bacteria not only modify surface elements to prevent adsorption and entry but also employ a wide array of antiphage strategies inside the cell. Antiphage systems are often carried on mobile genetic elements, which indicates that these islands can move en bloc from bacterium to bacterium via horizontal gene transfer (HGT) [9,38]. The role of these antiphage systems in phage resistance must be addressed in the development of phage therapeutics, especially durable off-the-shelf phage cocktails. The relationship of antiphage systems to phage resistance must be better understood so that phage therapeutics can be developed in a careful and rational manner to counter the myriad ways that bacterial pathogens become resistant. The natural presence of anti-antiphage systems in phages or their introduction via engineering could potentially be of great benefit in the development of more effective phage therapeutics.

To apply current knowledge of antiphage systems in the development of phage therapeutics, we utilized a web-based search tool known as PADLOC (prokaryotic antiviral defense locator) [39] for screening a panel of 100 highly diverse clinical isolates of *Pseudomonas aeruginosa*. The numbers of detected antiphage systems were correlated with susceptibility data for 70 phages; more phage-resistant strains carried significantly more antiphage systems. Some correlation was also found between phage resistance and the prevalence of certain antiphage systems.

## 2. Results

### 2.1. Identification of Antiphage Systems in the Genomes of 100 Diverse P. aeruginosa Clinical Isolates

We analyzed genome sequences from 100 highly diverse clinical isolates of *P. aeruginosa*. This diversity panel was selected based on core-genome multilocus sequence typing (MLST) from a large repository of multidrug-resistant isolates collected from across the U.S. Military Health System [40]. These isolates originated from multiple hospitals in the United States, as well as from Guam and Afghanistan, and include 91 distinct sequence types (STs). Genome sequence files deposited in the database at the National Center for Biotechnology Information (NCBI) were uploaded and analyzed using the PADLOC webserver tool [39].

Multiple antiphage systems were found in the *P. aeruginosa* strain panel, predicted based on sequence homology to a catalog of known defense systems. A total of 75 distinct antiphage systems were identified in the 100 strains (Figure 1). The number of systems reported does not reflect distinct subtypes (e.g., CBASS type I, II, III, and others), where there are multiple subtypes for many of the defense systems. Additionally, putative, nonexperimentally validated systems identified through “guilt-by-embedding” were identified in the diversity panel with the recent release of an updated catalog, PADLOC-DB v2.0.0 [41]. A total of 44 such phagedefense candidate (PDC) systems were identified after PADLOC analysis (Figure 2). While not all of these PDC genes may ultimately be confirmed as such, these results indicate areas for future discovery and the potential breadth of undiscovered antiphage systems in this bacterium. Overall, the average number of antiphage plus PDC systems in the 100 strains was 14.2.

Antiphage systems were not evenly distributed throughout the strains; many were found only in a small number of isolates, perhaps suggesting a relatively recent acquisition of these defense systems by *P. aeruginosa* strains via HGT. The average defense system was present in 13.25 genomes but ranged from 97 strains (PD-T4-6) to 1 strain (multiple systems). All but four of the identified systems were present in fewer than half of the diversity panel strains (Figure 1). Four antiphage systems were identified in 50 or more of the 100 strains: PD-T4-6, DMS-other, RM Type I, and SoFic. Thirty-seven systems were present in five or fewer strains. However, DMS-other is a catch-all grouping that reflects potential incomplete antiphage systems, new subtypes, or novel systems with known protein domains; thus, it should not be viewed as a distinct, broadly conserved, system. Since the diversity panel represents 91 different STs that were collected in different geographic locations over multiple years (Appendix A), the strains are likely to have acquired distinct sets of antiphage systems via HGT. One example to note is the distribution of PsyrTA, a toxin–antitoxin system that provides phage protection via abortive infection. PsyrTA was present in 11 strains that belong to four lineages, ST235, ST3002, ST3043, and ST2387.

The predicted antiphage systems provide defense in a variety of manners, including targeting phage nucleic acid invasion through its degradation, termination of infection by several different mechanisms (abortive lytic cycle), modification of phage DNA and RNA via a variety of epigenetic marks or incorporation of unusual bases, protein modification or degradation, production of RNA molecules that function in an RNAi-like manner, depletion of nutrients and cofactors, or physiological changes to the cell, among others that remain to be characterized. Some systems may have multiple mechanisms of action as early investigation into the Zorya defense system suggests [17]. Predicted diverse nucleases and abortive infection systems were the most prevalent among the 100-strain panel (Table 1). The PDC systems represent an additional group of putative antiphage systems for which characterization remains to be done. Some of these systems are reported to share sequence homology with other characterized systems. For example, PDC-S14 shares some overlap with GAO_29, a defense system that appears to act via a restriction-like mechanism [19].

Antiphage systems can be grouped by general mechanism into several types. For instance, 19 different major validated nucleic acid degradation or uncharacterized nuclease domain-containing systems were identified, not including subtypes. Eighteen types and multiple subtypes of abortive infection systems were identified. Other antiphage strategies included nucleic acid modification, formation of noncoding RNAs, epigenetic modification of invading phage nucleic acids, modification of proteins or altered protein activity, nutrient and cofactor depletion, or retardation of lysis. Many of these systems have only been recently identified, so more thorough characterization is still required. Some systems may display multiple mechanisms of action or have context-dependent mechanisms of action. The breadth of antiphage mechanisms demonstrates the numerous, overlapping, and complementary approaches that bacteria utilize to counter phage infection.

### 2.2. Phage Susceptibility of the Diverse P. aeruginosa Strains 

We determined the susceptibility of the 100 strains of *P. aeruginosa* to 70 phages that belong to 14 genera, *Pbunavirus*, *Nankokuvirus*, *Pakpunavirus*, *Phikzvirus*, *Yuavirus*, *Septimatrevirus*, *Epaquintavirus*, *Phikmvvirus*, *Pifdecavirus*, *Bruynoghevirus*, *Kochitakasuvirus*, *Litunavirus*, *Warsawvirus*, and *Hollowayvirus* (Appendix A). Two *P. aeruginosa* strains, MRSN 3587 and MRSN 8141, replaced previous members of the diversity panel after the host range data were collected for some phage–host pairs. Consequently, for those two strains, phage–host interaction data are available for only 44 phages. Plaque formation was used to determine the overall phage susceptibility percentage for each strain. Strains from the panel were susceptible to phages over a range of 0–90.0% (Figure 3a), and the average susceptibility value for the 100 strains was 38.7%. The number of antiphage systems encoded by each strain was plotted against the phage susceptibility percentage to address whether the presence of more antiphage systems is correlated with higher phage resistance. It was observed that, as strains encoded more antiphage systems, their resistance to phages increased (Figure 3a). The relationship between phage resistance and the presence of antiphage systems was statistically significant (*p* < 0.0001). The trend was weakly correlated with an R squared value of 0.2620, probably due to the role of other elements that mediate phage host tropism which includes surface receptors and RBPs, the presence or absence of prophages, and other physiological traits of the host strains. The diversity panel represents 91 STs of *P. aeruginosa*. Consequently, it was difficult to assess a potential correlation of ST with phage susceptibility, as most STs were represented by only one strain. One exception was ST235, a major, globally disseminated, lineage of concern [77]; six strains in the diversity panel belonged to this ST. Of these six isolates, five were phage-resistant, with an average susceptibility of 18.8%, which is significantly lower than the average susceptibility for the entire strain panel, 38.7% (Appendix A). The ST235 strains had an average of 16.5 antiphage systems per genome, which was slightly higher than that for the 100 strains, 14.2 systems (Appendix A). The structure of LPS, type-IV pilus, and other common phage receptors [78] has not yet been determined experimentally for the 100 strains used in this study. However, one can expect that the bacterial panel possesses diverse phage receptors since the analysis of *P. aeruginosa* isolates collected in the United States demonstrated their extensive diversity in O-antigen serotypes [79]. Our analysis of the 100 strains performed in this work using the web-based PAst tool [80] confirmed the diversity of O-antigen; 12 serotypes were identified in the panel (Appendix A). For some serotypes, the average susceptibility was lower or higher than the average of all strains, and this was correlated with the average number of antiphage systems. However, even within these groups, strain susceptibility and the number of antiphage systems fell over a wide range, with highly susceptible and highly resistant strains belonging to the same serotype. Also, the receptors for some of the phages used in this study are yet to be identified.

We also sought to identify a potential correlation of prophage content with *P. aeruginosa* phage susceptibility. To do so, the web-based PHASTER tool [81,82] was used for searching predicted prophage regions in the bacterial genomes. All 100 strains contained from one to fourteen prophage regions (Appendix A). There was a weak but statistically significant correlation between an increased number of prophages and a lower phage susceptibility. The number of prophage regions was also weakly correlated with the total number of antiphage systems (Appendix A). Interestingly, in the case of both serotype and prophage content, groups associated with phage resistance also tended to contain higher numbers of antiphage systems. The factors influencing phage susceptibility are thus multifaceted and require deeper analysis. 

There were clear outlier strains, which might make for an interesting follow-up analysis. Of interest are strains like MRSN 317, which possesses only six antiphage systems and yet is lysed by only 15% of the phages tested (Appendix A). Other strains that exhibit very low susceptibility to the phages but have relatively few identified antiphage systems include MRSN 6220, MRSN 6678, and MRSN 13488. These are among the 10 most phage-resistant strains in the panel (<3% phages show productive infection) and yet only possess 12, 13, or 9 antiphage systems, respectively. Outliers that were more phage-susceptible even with many predicted antiphage systems included strains MRSN 11976 and MRSN 1899. These strains possess 26 and 18 predicted antiphage systems, respectively; MRSN 11976 and MRSN 1899 are lysed by 63% and 68% of the phages tested, respectively (Appendix A). The average number of antiphage systems possessed by the most resistant strains was 19.2 (Table 2). That means that these highly phage-susceptible strains had a number of defense systems that were equivalent or above the average for the most resistant strains.

Antiphage systems may not fully prevent productive lysis, but instead reduce the efficiency of plating (EOP), so the impact of antiphage systems on the efficiency of plating was assessed (Figure 3b). Phage titers were normalized to the highest value for that phage on the 100-strain panel. For any strain, all EOP values of the 70 phages were then averaged. This yielded an average EOP ranging from 0 (MRSN 6678) to 0.26 (MRSN 11281) (Appendix A). The results showed similar trends to those observed for the presence or absence of plaque formation. The presence of higher numbers of antiphage systems was correlated with a lower average EOP of phages on that strain (Figure 3b). This trend was statistically significant (*p* < 0.0001) but with a weak correlation (R squared = 0.2058). The relationship of antiphage systems with average EOP and phage susceptibility together was then analyzed. A general trend was observed that, as strains had more antiphage systems, both phage susceptibility and EOP were lower (Appendix A). Along with the correlation of the antiphage systems prevalence with phage resistance and low EOP, there were outliers that may represent interesting follow-up with an investigation into the receptor structure, role of prophages, or assessment of the activity of predicted antiphage systems. For example, MRSN 435288 was predicted to encode only five antiphage systems; yet, the 70 phages plated on this strain have a very low average EOP of 0.0037 (Appendix A). 

We hypothesized that certain antiphage systems could be particularly important in driving phage resistance. Then, it was determined if some antiphage systems are more prevalent in phage-resistant strains and if some are relatively rare. The median point in phage susceptibility was approximately 34%. The 100 strains were divided into four separate groups: most resistant (lysed by <3%); intermediate resistant (lysed by 3–35%); intermediate susceptible (lysed by 35–75.5%); and most susceptible (lysed by >75.5%) (Appendix A). The most resistant and intermediate resistant strains possessed an average of 19.2 and 16.8 antiphage systems per strain, respectively. Conversely, intermediate susceptible and most susceptible strains had 11.9 and 7.6 antiphage systems per strain (Table 2).

Altogether, 22 validated antiphage systems and three PDC systems were present in 15 or more of the *P. aeruginosa* strains. An antiphage system was considered prevalent in a phage-resistant strain if it represented 52.5% or greater of the total strains in the panel encoding that particular system. Of the 22 validated systems, 19 were prevalent in the phage-resistant strains (Figure 4). The prevalence ranged from 86.7% phage resistance in the strains encoding Zorya (*n* = 15) to 53.8% resistance in the strains encoding Lamassu Family systems (*n* = 26). One of the three remaining systems, PD-T4-6, was intermediate, with no prevalence in the resistant or susceptible strains (Appendix A). Two other systems, CRISPR–Cas with all Cas types and DRT of all types, were prevalent in susceptible strains, with 53.5% of the 43 strains encoding CRSIPR–Cas systems and 56% of the 25 strains encoding DRT in the susceptible groups.

Other defense systems were also prevalent in resistant strains but were less represented within the 100-strain panel. These included RosmerTA (69.2% in resistant strains), PsyrTA (81.8% in resistant strains), and retrons (54.6% in resistant strains) (Appendix A). Some antiphage systems showed no prevalence in resistant or susceptible strains, including BREX and Septu, which showed a 50/50 split (both systems are predicted to be encoded by 14 strains). A few rare systems showed relative prevalence in susceptible strains, including 10 strains encoding PD-Lambda systems (60% of them were phage susceptible), seven strains encoding AVAST (71% of them were susceptible), and nine strains encoding Mza (56% of them were susceptible) (see Appendix A). 

The three PDC systems found in 15 or more strains were prevalent in resistant strains. PDC-S02 was predicted in 39 strains, of which 59.0% were resistant. PDC-S06 was predicted in 38 strains, of which 74.4% were resistant. Lastly, PDC-M30 was predicted in 32 strains, of which 68.8% were resistant. Other, more rarely occurring PDC systems, were found at a higher frequency in resistant or susceptible groups. For example, PDC-S09 and PDC-S11 showed strong prevalence in resistant strains, with 82% and 90% of 11 and 10 strains encoding them, respectively. One system, PDC-S08, showed an even 50/50 split, and PDC-S14 showed spread in susceptible strains (57% of 14 strains; Appendix A).

## 3. Discussion

This study provides a snapshot of the distribution of the currently identifiable antiphage systems in a highly diverse panel of 100 *P. aeruginosa* strains, and how the presence of these antiphage systems relates to susceptibility to a collection of 70 phages that includes 14 genera, comprising myo-, sipho-, and podophages. These diverse panels of bacterial strains and phages provide both a significant sample of the defense systems present in *P. aeruginosa* and an indication of how effective they are against the broader diversity of *Pseudomonas* phages. This effort further supplies a snapshot of the prophage content in the 100 diverse strains. We observed a weak, but statistically significant correlation between the number of antiphage systems and the number of prophage regions (Appendix A). This was not surprising as prophages are known hotspots of antiphage systems, often to furnish superinfection immunity [8,9]. On the other hand, the O-antigen serotype was not correlated with antiphage systems or phage susceptibility. *P. aeruginosa* strains belonging to the same serotype displayed broadly divergent antiphage systems content and phage susceptibility (Appendix A). This is likely due to horizontal gene transfer as antiphage systems are frequently found on mobile genetic elements [9]. 

Understanding the representation and frequency of different antiphage systems in *P. aeruginosa* clinical isolates can provide critical information for the development of rationally designed phage therapeutic cocktails. This knowledge can empower the choice of therapeutic phages that counter these defense systems, either by evasion or by the production of dedicated antidefense enzymes [28,29,83]. Coupling this new information with the current standards of phage selection, including safe genomic properties, broad host range, robust lytic and antibiofilm activity, using different receptors, and confirmation of synergy with other phages and antibiotics can enable the rational design of more effective and durable therapeutic phage cocktails [31]. As phage engineering approaches also continue to improve [84], the incorporation of anti-antiphage genes that counter identified antiphage systems into candidate therapeutic phages could improve their efficacy and durability as therapeutics. 

An initial avenue of possible utilization of these results is the identification of candidate phages encoding antidefense genes. Some of these genes have been identified and characterized [29,85]. In this work, we identified some phages that broadly lyse strains encoding certain antiphage systems (e.g., pycsar), for which known antidefense genes have been identified. For example, phage KEN5 lyses 7/8 strains encoding a pycsar effector, while KEN3 is able to lyse 6/8 of these strains. These strains are genetically diverse and belong to seven different STs, indicating that these phages may be broadly active against more diverse *P. aeruginosa* strains. While the extensive information on LPS types and RBPs is currently unavailable, it seems clear that if the putative pycsar genes are active, these two phages are somehow unaffected by the system that is prevalent in phage-resistant strains (62.5% of strains with pycsar are resistant) (Appendix A). This dataset could also provide initial information for identifying novel antidefense enzymes. Shango has been discovered and characterized in *Escherichia coli* and *P. aeruginosa* [5,7], but no phage-encoded anti-Shango enzymes have yet been identified. A *Pbunavirus* phage EPa11 can lyse 5/7 strains encoding putative Shango systems (Appendix A). If the predicted Shango systems are active in these strains, it suggests that this phage is either somehow unaffected by or can counter this defense system. While these data alone are insufficient to establish that novel antidefense enzymes are present, these phages are certainly candidates for the discovery of new antidefense systems. Further discovery of these systems can be leveraged to suppress the emergence of phage resistance in the design of improved phage therapeutics.

In addition, our analysis could provide information on the types of phages that are not well targeted by certain defense systems, a phenomenon that has been observed for multiple antiphage systems [19]. With respect to our pycsar example, KEN3 is a podovirus in the genus *Bruynoghevirus*, while KEN5 is a myovirus in the genus *Pakpunavirus* (Appendix A). A broader analysis of phage–host interaction data could potentially define preferences for antiphage systems related to either taxonomic or structural factors. These are important avenues of future investigation and require further in silico analysis and laboratory work.

The diversity in the types of antiphage systems that were identified in the 100-strain *P. aeruginosa* panel is remarkable. Most of the strains encoded both an abortive infection strategy and a nucleic acid degradation mechanism. Substantial numbers of strains encoded more unique systems with a variety of mechanisms, including nucleic acid modification and protein modification or degradation. Countering different steps of phage infection with multiple and different antiphage systems likely contributes to phage resistance in a synergistic manner. Such synergy has been observed for many antiphage systems, including those that seem mutually exclusive per comparative genomic analysis [86]. The coupling of nucleic acid-degrading, phage gene expression control, and cell suicide mechanisms may represent a complementary array of antiphage mechanisms that provide flexibility in the speed of response and cost to the bacterial cell [87]. In our analysis, CRISPR–Cas systems were not prevalent in the phage-resistant strains. This paradoxical outcome is in agreement with recent data of Lood et al. [81], who observed that strains of *P. aeruginosa* encoding CRISPR–Cas systems were more susceptible to a panel of 14 phages than strains lacking CRISPR–Cas. More work is clearly needed to understand synergies and antagonisms among antiphage systems.

In the PADLOC analysis used here (based on sequence homology), many antiphage systems that are multigene (located in operons) were readily identified. However, there are also single-gene antiphage systems, including SoFic, some PD systems (e.g., PD-T4-6), and others that are more subject to false calls. We have not independently assessed whether these systems are expressed in a given strain, and, if so, whether they play any role in phage defense. The number of strains, systems, and possibility for redundancy makes such analysis very time- and labor-consuming. This can be even more important to assess if some systems may have roles in the bacterial cell beyond phage defense. For example, wadjet systems have been reported to provide protection from exogenous DNA, including not only phage DNA but also plasmids, transposons, or other mobile genetic elements [43]. Some wadjet systems were shown to be involved in phage defense, but some may only be activated in response to certain signals or phage components. Understanding the context of defense system activation is important. Even growth conditions in the laboratory could impact their expression. For example, in one *P. aeruginosa* strain, CRISPR–Cas type I-F was under the control of a two-component system involved in regulating alginate biosynthesis. Phages hijacked a repressor of this two-component system to silence the expression of the CRISPR–Cas genes [88]. While we have an initial picture of antiphage machinery in the genomes of diverse *P. aeruginosa* strains, whether, how, and when these systems respond to phage infection is yet to be elucidated. As antiphage systems continue to be identified and characterized, and their role in phage–host interactions is established, this picture will continue to become clearer.

To find if there was a correlation between the number of antiphage systems and phage resistance, we first relied on the presence or absence of plaque formation. This provides data as to if a strain is resistant under laboratory conditions but may lead to an underappreciation of “soft” resistance. Where antiphage systems have been characterized, they frequently reduce the efficiency of plating (EOP), without eliminating lytic activity. For example, the PD-T4-6 system identified by Vassallo and colleagues conferred an approximately four-log reduction in EOP for phage T4, but plaques were still formed [10]. Consequently, we calculated the average EOP for the panel of 70 phages on each of the *P. aeruginosa* diversity-panel strains to provide an approximation of how well phages plate on each strain. EOP was correlated with the number of antiphage systems predicted by the PADLOC analysis, and some protection was revealed in the form of reduced EOP. However, numerous factors can influence the EOP, including the presence of primary or secondary phage receptors, characteristics of the strains, or even the presence or absence of certain plasmids [89,90]. To empirically assess whether the presence of a system affects EOP, it would be necessary to conduct a comparative EOP analysis with antiphage systems knocked out in isogenic strains. However, given the diversity and number of antiphage systems, strains, and phages, vast resources exceeding our current capacity would be needed to carefully assess synergy in the analyzed strains beyond the marker that was selected.

While we found a statistically significant correlation between the number of antiphage systems and phage resistance in *P. aeruginosa* strains, this trend may not be applicable to all bacterial species. For instance, a recent analysis of *E. coli* phage–host interactions found no relation between antiphage systems and phage susceptibility [91]. Phages of *P. aeruginosa* and other gram-negative bacteria often adsorb to LPS, a molecule with complex structure and high diversity, particularly in the O-antigen [78]. There are twenty characterized O-antigen serotypes within *P. aeruginosa* [92], while *E. coli* has a much greater diversity, with approximately 180 O-antigen serotypes [93]. Strains from our panel represent 12 O-antigen serotypes (Appendix A). The restrictions that higher receptor diversity imposes on host range may reduce the need to maintain a diverse arsenal of antiphage systems. For a species with less diversity in common phage receptors, more antiphage systems may be necessary to provide adequate protection against infection to be successful in a particular niche. Despite the lower number of antiphage systems in *E. coli*, Gaborieau and colleagues were able to identify a weak but statistically significant correlation between antiphage systems and reduced viral infectivity [91]. In species with a greater diversity of phage receptors, antiphage systems may play a more secondary role by reducing the infectious efficacy of the phages that adsorb to the bacterial cell. Consequently, an analysis of antiphage systems in phage–host dynamics should consider all of the elements that drive phage–host tropism in a particular bacterium.

## 4. Materials and Methods

### 4.1. Bacterial Strains Used in This Study

The 100 *P. aeruginosa* clinical isolates used in this study were provided by the Multidrug-resistant organism Repository and Surveillance Network (MRSN), which developed this panel to maximize genetic diversity [40]. The panel is available at BEI Resources (https://www.beiresources.org), Catalog #NR-51829. 

### 4.2. Phages Used in This Work

Phages used in this study represent a diverse group of 50 unique lytic *Pseudomonas* phages isolated from sewage and environmental waters over almost 10 years at the Walter Reed Army Institute of Research (WRAIR), as well as 20 unique phages isolated by WRAIR’s overseas laboratories in Kenya at the U.S. Army Medical Research Directorate—Africa (KEN phages) and Thailand at the Armed Forces Research Institute of Medical Sciences (AFRIMS, AFR phages).

### 4.3. Handling of Bacterial Cultures and Phages

All strains were cultured overnight in Heart Infusion Broth (HIB, BD, Franklin Lakes, NJ, USA) at 37 °C with shaking prior to use for host range testing. All phages were purified from environmental samples following our standard approach [94]. Phages were stored in a propagation medium (HIB + 0.1% glucose, 2 mM MgCl_2_, 0.5 mM CaCl_2_) at 4 °C until the use in host range assays. 

### 4.4. Analysis of Bacterial Genomes for Identification of Antiphage Systems

We utilized the PADLOC webserver, https://padloc.otago.ac.nz/padloc/ (accessed on 27 October 2023), employing their full catalog of defense systems when collecting search results [39]. This means that antiphage systems not fully characterized or experimentally validated, or reported in pre-prints, were included. We wanted to obtain the broadest representation possible of antiphage systems present in the *P. aeruginosa* diversity panel and thus did not exclude these predicted putative systems. Genome sequences of the 100 *P. aeruginosa* panel strains available in the National Center for Biotechnology Information (NCBI) database [40] were used as queries for the webserver search tool, and the results were collected following a run with CRISPRdetect analysis. The results were analyzed, such that the total numbers of strains encoding each system were determined. 

### 4.5. Analysis of Bacterial Genomes to Identify Serotype

We utilized the PAst webserver [80], https://cge.food.dtu.dk/services/PAst/ (accessed on 11 January 2024) to assign a serotype to our diversity panel strains. FNA files for each strain were uploaded and run to provide a predicted serotype for each strain. The strains were grouped by serotype and the average phage susceptibility and average number of antiphage systems were determined for every group.

### 4.6. Analysis of Bacterial Genomes to Identify Prophage Content

We utilized the PHASTER webserver [81,82], https://phaster.ca/ (accessed on 12 January 2024) to screen bacterial genomes for prophage regions. The prophage content in each strain was determined, and the total number of prophage regions (including predicted intact, incomplete, and questionable prophages) was tallied for every strain. The total number of prophage regions was compared with phage susceptibility and the number of antiphage systems.

### 4.7. Phage Susceptibility Testing

Phage susceptibility was assessed using a micro-spot plaque assay [95], where phage dilution series are plated on all 100 strains of the panel, and plaque formation was monitored. ”Lysis from without” or a nonreplicative lysis was considered as a negative result (phage resistance). 

### 4.8. Determination of Phage Susceptibility Groups

To provide a breakdown of strain susceptibility to the phages, a division at the median phage susceptibility was made, where 50 strains based on phage susceptibility calculations were considered more resistant and the other 50 strains more susceptible. Within the halves, the 10 most resistant and 10 most susceptible strains were placed in groups. The number of antiphage systems present in each strain within each group was averaged to yield the average number of antiphage systems present within that group. 

### 4.9. Correlation of Antiphage Systems with Phage Susceptibility

To address the core question of how an antiphage-system genome relates to phage susceptibility, a plot comparing these datasets was generated, and an XY correlation analysis was performed to assess the statistical significance using the available statistical analysis tools from GraphPad Prism (version 9.5.1, Dotmatics, Boston, MA, USA).

### 4.10. Correlation of Antiphage Systems with Average Efficiency of Plating

To account for the possibility of partial antiphage activity that does not fully prevent productive lysis, we determined the average EOP for each strain. To do so, each phage’s titer was normalized to the highest titer observed on any of the 100 strains. For phage–strain interactions with no plaque formation or nonproductive lysis, these were considered an efficiency of 0. The EOPs for all 70 phages on each strain were averaged together. The calculated average EOP values were plated against the number of antiphage systems, and an XY correlation analysis was run to assess statistical significance (GraphPad Prism 9.5.1). 

### 4.11. Assessment of Prevalence of Antiphage Systems in Phage Susceptibility Groups

The distribution of strains encoding each antiphage system in a phage susceptibility group was determined by reviewing the PADLOC results. To determine if a particular system was prevalent in resistant or susceptible strains, the distribution of antiphage systems was assessed across the susceptibility groups (most resistant, intermediate resistant, intermediate susceptible, and most susceptible). If >52.5% of strains encoding a system belonged to resistant or susceptible groups, that antiphage system was considered prevalent in either resistant or susceptible strains. 

## 5. Conclusions

Phages have coevolved with their bacterial hosts for billions of years, in a very long evolutionary process (that likely preceded the divergence of the bacteria from the rest of life). The host cell has developed various means to avoid being infected with these viruses, as the viruses, in turn, develop means to overcome or circumvent these defenses. Complex and multilayered phage–bacteria interactions continue to be unraveled in the current renaissance of phage research. Bacterial resistance to phages represents a major and persistent challenge in the development of efficacious phage therapeutics, as has been experienced in the antibiotic saga. Efforts to subvert or prevent phage resistance in therapeutic applications have largely focused on designing multiphage cocktails that target multiple receptors with complementary and overlapping component host ranges. These efforts are important and should continue toward developing effective and durable phage therapeutics, but, in this endeavor, attention must be paid to addressing bacterial defense strategies. While much remains to be discovered, early analysis is already showing that certain antiphage systems have clear specificities for the types of phages they target [19]. With a better understanding of the types of antiphage mechanisms present within a species of interest, such as *P. aeruginosa*, rational decisions can be made to select therapeutic phages that are less subject to, or that can evade or counter, prevalent antiphage systems. To our knowledge, this is the first work relating the in silico analysis of antiphage-system genome content using PADLOC with a broad analysis of the susceptibility of a highly diverse panel of host strains to a diverse collection of phages. Our results indicate that the multiple and varied antiphage systems present in the genomes of diverse *P. aeruginosa* clinical isolates have distinct patterns of association with phage resistance and susceptibility and, thus, are likely to play an important role in mediating phage resistance. Understanding these interactions is critical for the future development of phage-based antibacterial drugs. 

## Figures and Tables

**Figure 1 ijms-25-01424-f001:**
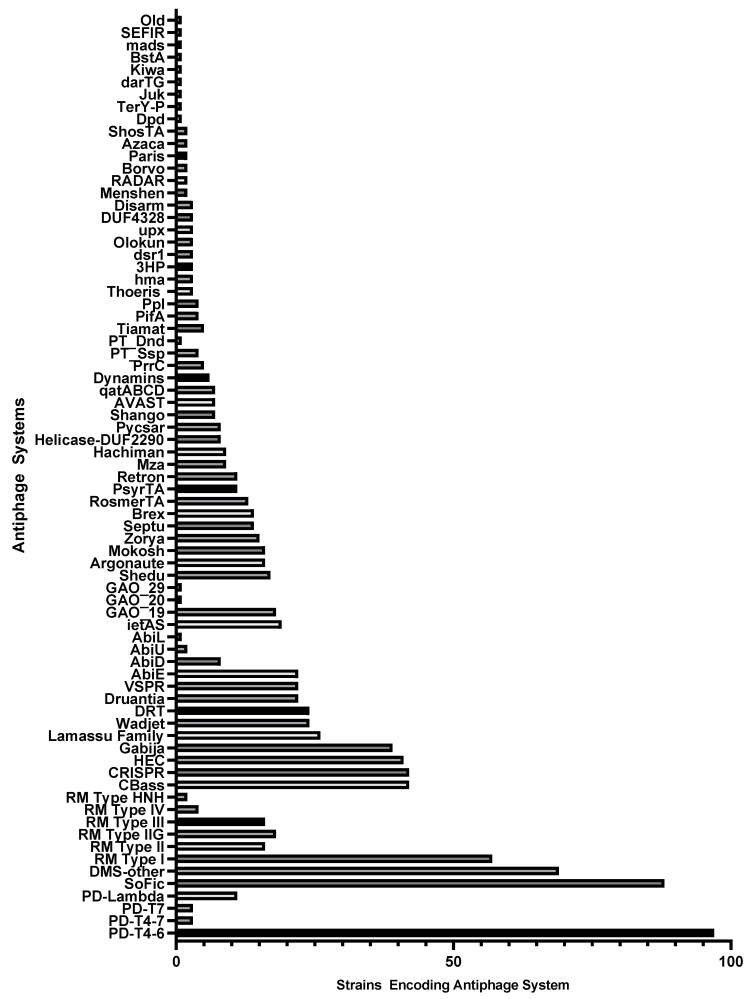
Antiphage systems identified in a 100-strain diversity panel of *P. aeruginosa* and the number of strains encoding each system.

**Figure 2 ijms-25-01424-f002:**
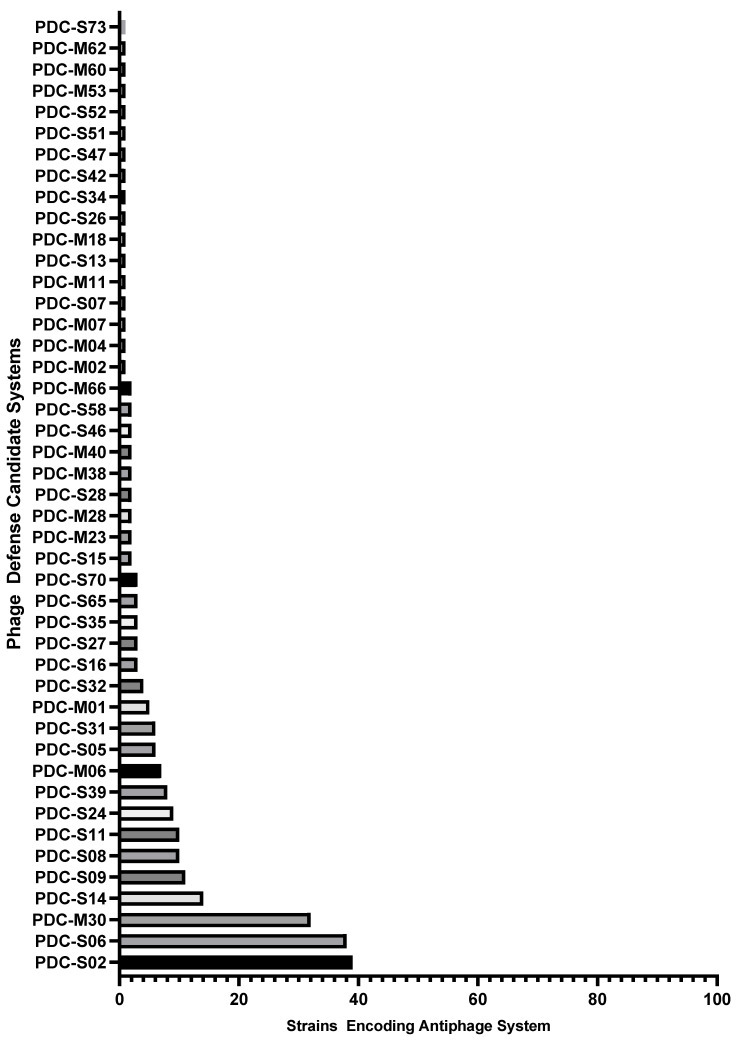
Phage defense candidates (PDCs) identified by a “guilt-by-embedding” approach and the number of strains from the diversity panel encoding each system.

**Figure 3 ijms-25-01424-f003:**
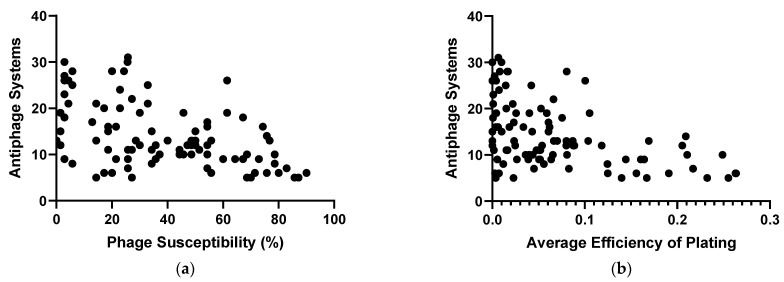
(**a**) Susceptibility of *P. aeruginosa* strains to a panel of phages plotted against the number of antiphage systems identified in each strain via PADLOC analysis. (**b**) Average efficiency of plating for a panel of phages plotted against the number of antiphage systems.

**Figure 4 ijms-25-01424-f004:**
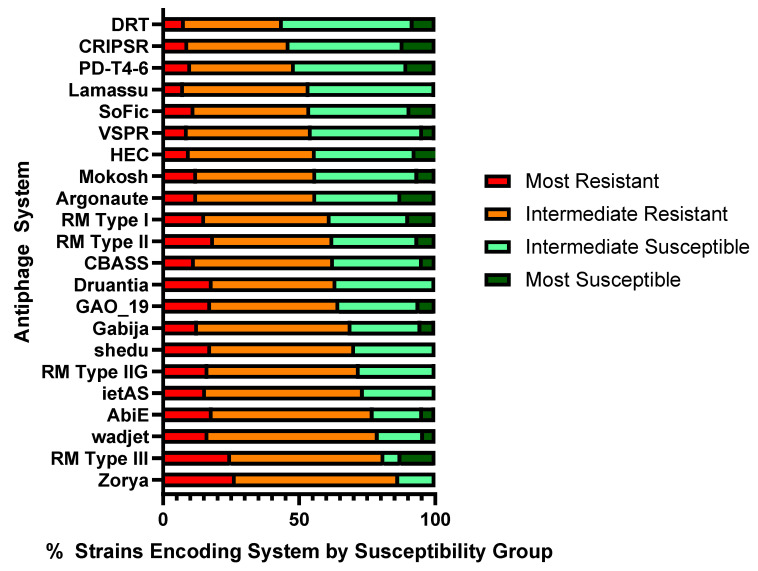
Antiphage systems present in 15 or more strains, represented by percent distribution in phage susceptibility groups. If greater than 52.5% of the strains encoding a system are categorized as resistant, the system is considered prevalent among phage-resistant strains.

**Table 1 ijms-25-01424-t001:** Experimentally validated or predicted mechanism of antiphage activity for identified systems in the diversity panel.

Mechanism of Antiphage Action	System	Number of Strains Encoding	References
Nuclease Activity—Nucleic Acid Degradation	R-M systems	67	[18]
	CRISPR–cas systems *	42	[42]
	Gabija	39	[17]
	Wadjet	24	[17,43]
	Druantia	22	[14,17]
	Shedu	17	[44]
	Mokosh	16	[7]
	Zorya *	15	[17]
	Septu	14	[14,17]
	Mza	9	[19]
	qatABCD	7	[14,19]
	Ppl	4	[45]
	Olokun	4	[7]
	DISARM	3	[46]
	Upx	3	[19]
	Menshen	2	[7,47]
	Azaca	2	[7]
	Kiwa	1	[17,48]
	Old	1	[8]
Abortive Infection or Cell Dormancy	PD Systems (T4-6, T4-7, T7, λ)	100	[10]
	Abi systems (E, D, U, L)	30	[49,50,51,52]
	GAO_19	18	[19]
	CBASS	42	[53]
	Lamassu	26	[7]
	ietAS	19	[19,26]
	RosmerTA	13	[7,54]
	PsyrTA	11	[7,55]
	Hachiman	9	[17,26,56]
	Helicase–DUF2290	8	[8]
	Pycsar	8	[57]
	AVAST	7	[19,58]
	PrrC	5	[59,60]
	PifA	4	[61]
	Paris	2	[8]
	ShosTA	2	[7]
	darTG	1	[55,62]
	BstA	1	[63]
Nucleic Acids Modification/Reverse Transcriptases/Expression Modification	DRT	24	[19]
	Argonaute	16	[64,65,66]
	BREX	14	[67]
	Retrons	11	[68,69]
	PT_Ssp and PT_Dnd	5	[70]
	RADAR	2	[19]
	Dpd	1	[71]
	MADS	1	[72]
Protein Modification	SoFic	88	[7]
	Borvo	2	[7]
	TerY-P	1	[19]
Other	Dynamins (Lysis delayance)	6	[73]
	Thoeris (NAD+ depletion)	3	[74]
	Dsr (NAD+ depletion)	3	[21]
	SEFIR (NAD+ depletion)	1	[7]
Unknown	DMS other	69	[39]
	HEC (Hma embedded candidates)	41	[39]
	Shango	7	[7]
	Tiamat	5	[7]
	Hma	3	[56]
	3HP	3	[8]
	DUF4238	3	[8]
	Juk	1	[75]

* Multiple subtypes are described for some defense systems. In analyzing distribution, subtypes were grouped together. Zorya and some CRISPR–Cas proteins may have multiple mechanisms of action both degrading invading phage DNA and inducing dormancy or cell death [17,76].

**Table 2 ijms-25-01424-t002:** Average number of antiphage systems present per strain in different phage susceptibility groups.

Phage Susceptibility Group	Average Antiphage Systems/Strain
Most resistant 10%	19.2
Intermediate resistant	16.8
Intermediate susceptible	11.9
Most susceptible 10%	7.6

## Data Availability

Data are contained within the article and Appendix A.

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
