# Peer review of "Correlation of Pseudomonas aeruginosa Phage Resistance with the Numbers and Types of Antiphage Systems"

_ijms, 2024, doi:10.3390/ijms25031424_

Round 1

Reviewer 1 Report

Comments and Suggestions for Authors

The manuscript «Correlation of Pseudomonas aeruginosa phage resistance with numbers and types of antiphage systems» is devoted to the study of the correlation between the presence of various antiphage systems in Pseudomonas aeruginosa and their resistance to bacteriophages using modern molecular and bioinformatic methods. The authors analyzed a collection of 100 Pseudomonas aeruginosa clinical strains and 70 different isolates of bacteriophages specific to Pseudomonas aeruginosa. In the study, using genomic databases, a wide range of antiphage systems was identified in Pseudomonas aeruginosa using the PADLOC program. Undoubtedly, bacterial protection systems are extremely complex, diverse, changeable and have not yet been fully studied.

Overall, this is a well-written and well-designed article. The data are interesting. I think, this manuscript is suitable to publish in this Journal.

Comments:

1. Undoubtedly, the manuscript can be improved by studying for the presence of prophages, and differences in cellular receptors in the studied strains of Pseudomonas aeruginosa and the connection of these data with the resistance of Pseudomonas aeruginosa strains to lysis by bacteriophages.

2. The authors have repeatedly written about the importance of this study for antibacterial therapy. However, the authors do not provide data on which specific isolates of phages with a wide spectrum of antibacterial action could be promising for the design of phage therapeutic cocktails.

Author Response

Thank you very much for reviewing our manuscript ijms-2798400 (“Correlation of Pseudomonas aeruginosa phage resistance with numbers and types of antiphage systems”) submitted to IJMS and your important comments that helped us to improve the manuscript. Please see below our point-by-point responses to your comments in red. We agreed with all the comments and made appropriate edits in the manuscript and provide the revised text with markup and the clean text.

 Reviewer 1 Comments:

  1. Undoubtedly, the manuscript can be improved by studying for the presence of prophages, and differences in cellular receptors in the studied strains of Pseudomonas aeruginosa and the connection of these data with the resistance of Pseudomonas aeruginosa strains to lysis by bacteriophages.

We fully agree with the Reviewer. Following this comment, we analyzed the number of prophages in each of the 100 P. aeruginosa genomes using the web-based tool PHASTER (please see the revised marked up manuscript and the newly added Supplementary Table S5 and Supplementary Figure S1). We also concur that the identification of phage receptors would help to find out if phage resistance is receptor-dependent or some antiphage systems might be involved. However, even adsorption tests of 70 phages against 100 strains would include testing thousands of combinations, which is a gigantic work. Thank you for this comment that shapes our future studies. At this point, we addressed this comment by analyzing O-antigen types of the 100 strains of P. aeruginosa using the web-based PAst tool, because it is well known that O-antigen and other parts of LPS are frequently used by P. aeruginosa as receptors. Please see the results in the revised manuscript and newly added Supplementary Table S4.

  1. The authors have repeatedly written about the importance of this study for antibacterial therapy. However, the authors do not provide data on which specific isolates of phages with a wide spectrum of antibacterial action could be promising for the design of phage therapeutic cocktails.

This manuscript was focused on P. aeruginosa antiphage systems and phage resistance. We are working on two additional manuscripts focused on host ranges, genomic properties, compatibility, other characteristics of P. aeruginosa phages, and their combination into therapeutic cocktails and preclinical testing in animal models. However, there is information on the breadth of phage activity in Table S7. For example, phages with broad host ranges promising for therapeutic use are: AFR26, AFR28, AFR40, KEN5, EPa15, EPa22, and others.

Sincerely,

Andrey Filippov, Ph.D., correspondent author

Reviewer 2 Report

Comments and Suggestions for Authors

Correlation of Pseudomonas aeruginosa phage resistance with 2 numbers and types of antiphage systems”

The authors analysed antiphage systems using PADLOC web-server tool with 100 P. aeruginosa isolates and 50 phages. They showed that phage resistance and the presence of antiphage systems in bacterial chromosomes were correlated for the first time.

This report would be helpful for phage scientists to establish efficient phage therapeutic strategy against P. aeruginosa infections even though many PDC should be studied in depth for the functional roles. First barrier for phages to infect host bacteria would be phage adsorption via receptors and second would be antiphage systems. If the authors use phages which are adsorb via same receptor, the correlation between phage resistance and the presence of antiphage systems in bacteria would show better statistical outputs.

Did you analyse the presence of antiphage systems according to sequence types (STs)? 

 "P. aeruginosa" name should be italic in the whole manuscript

Author Response

Thank you very much for reviewing our manuscript ijms-2798400 (“Correlation of Pseudomonas aeruginosa phage resistance with numbers and types of antiphage systems”) submitted to IJMS and your important comments that helped us to improve the manuscript. Please see below our point-by-point responses to your comments. We agreed with all the comments and made appropriate edits in the manuscript and provide the revised text with markup and the clean text.

Reviewer 2 Comments:

  1. The authors analysed antiphage systems using PADLOC web-server tool with 100 P. aeruginosa isolates and 50 phages. They showed that phage resistance and the presence of antiphage systems in bacterial chromosomes were correlated for the first time.

Thank you for this comment. Actually, we used 70 phages in our analysis.

  1. This report would be helpful for phage scientists to establish efficient phage therapeutic strategy against P. aeruginosa infections even though many PDC should be studied in depth for the functional roles.

We appreciate the positive evaluation of our results and agree that characterization of PDC systems for their functions is important and will add a lot of knowledge to phage-bacteria interactions.

  1. First barrier for phages to infect host bacteria would be phage adsorption via receptors and second would be antiphage systems. If the authors use phages which are adsorb via same receptor, the correlation between phage resistance and the presence of antiphage systems in bacteria would show better statistical outputs.

We 100% agree with the Reviewer that identification of receptors or at least adsorption tests would be very important to figure out if phage resistance is receptor-dependent or some antiphage systems might be involved. However, even adsorption assays of 70 phages against 100 strains would include testing thousands of combinations. This is a gigantic work. Thank you for this comment that shapes our future studies. At this point, we addressed this comment by analyzing O-antigen types of the 100 strains of P. aeruginosa using the web-based PAst tool, because it is well known that O-antigen and other parts of LPS are frequently used by P. aeruginosa as receptors. Please see the results in the revised (marked up) manuscript and newly added Supplementary Table S4.

  1. Did you analyse the presence of antiphage systems according to sequence types (STs)? 

We had not done it before. We did it now, following your comment. We found some potential correlation of ST235 with phage resistance and more antiphage systems in the genomes (please see the revised manuscript and new Supplementary Table S3). Thank you for your question!

  1. "P. aeruginosa" name should be italic in the whole manuscript.

It was italicized in the whole manuscript, but the manuscript file was modified after submission to IJMS and some binomials and phage genera were de-italicized. Now we are making sure that “P. aeruginosa” is in italic everywhere in the text.

Sincerely,

Andrey Filippov, Ph.D., correspondent author